# OpenReview forum: "Echoing: Identity Failures when LLM Agents Talk to Each Other"
_ICLR.cc/2026/Conference — ICLR 2026 Conference Desk Rejected Submission_

### Official Review · Reviewer_f5W6 · 2025-10-31

**Soundness:** 3
**Presentation:** 2
**Contribution:** 3
**Rating:** 6
**Confidence:** 3

**Summary:**

This paper investigates a failure mode in agent-vs-agent (AxA) dialogues, termed echoing, in which an agent abandons its assigned identity and adopts that of its counterpart. The authors formalize AxA interactions and conduct experiments across different LLM backbones, prompts, and domains (e.g., hotel booking, car sales, and supply chain). Experimental results show that echoing is prevalent (occurring in approximately 5%–70% of cases), even in “reasoning” models (averaging about 32.8%), and often emerges in the later stages of dialogue (typically after seven turns), frequently masked by standard completion metrics. Furthermore, the authors propose a protocol-level scheme that mitigates echoing through structured responses, which reduces but does not completely eliminate it.

**Strengths:**

(1) This paper formally defines echoing as a distinct AxA failure, distinguishing it from generic errors or hallucinations as well as from prior MAS settings, and clearly elucidates the AxA framework.

(2) This paper presents extensive experiments across multiple LLMs and domains, providing a solid empirical foundation for echoing research. Furthermore, the study analyzes failed cases and extracts relevant principles (prevention, non-intrusion, seamlessness, and architectural integration).

(3) This study demonstrates that echoing persists even in stronger “reasoning” models and that standard evaluation metrics often mask this phenomenon. These findings prompt a rethinking of evaluation methods for AxA systems.

**Weaknesses:**

(1) The text in the third section of Figure 1 is too small to be easily read.

(2) The three domains used in the experiments are all transactional and relatively structured. It is recommended to add at least one less-structured domain to examine whether echoing is a common AxA phenomenon beyond task-oriented setups.

**Questions:**

Please See Weaknesses

---

> ### Author Response · Authors · 2025-11-21
> **Review response**
>
> We thank the reviewer for recognizing that our paper (1) "formally defines echoing as a distinct AxA failure," (2) presents "extensive experiments... providing a solid empirical foundation," and (3) demonstrates that "echoing persists even in stronger 'reasoning' models" with "standard evaluation metrics often mask this phenomenon." We are particularly encouraged by the recognition of our contribution and the observation that our findings "prompt a rethinking of evaluation methods for AxA systems."
>
> ## Responses to Weaknesses
>
> **W1: Figure 1 text too small**
>
> We have increased the font size in Figure 1 to improve readability in the revised version. Thank you for the feedback.
>
> **W2: Three domains are transactional/structured; add less-structured domain**
>
> We appreciate this observation. Our focus on transactional, negotiation-oriented AxA scenarios was deliberate:
>
> **Practical Relevance**: These domains represent the most immediate real-world deployment contexts for AxA systems. Multiple industry/academic efforts (Google's A2A protocol \[54\], IBM's ACP \[47\], Cisco's AGNTCY.org \[61\], and others \[41, 68\]) are specifically targeting transactional agent-to-agent interactions for enterprise applications.
>
> **Pragmatic Focus on Agent-as-Consumer**: In typical agent-human settings, today’s agent almost universally assumes the enterprise/business role, acting as an assistant, seller, or service provider (e.g., customer service bot, booking agent). Conversely, the human is the "consumer." We strongly suspect alignment procedures optimized for this dominant setting and thus introduces a strong bias that make the agent-as-consumer role inherently susceptible to identity drift when interacting with a business-role agent.
>
> **Identity Boundaries are Clearest**: Transactional domains provide well-defined, distinct agent identities (buyer vs. seller, customer vs. provider) that make echoing behavior unambiguous to detect and measure, allowing us to establish a reliable baseline.
>
> **Echoing Severity**: We hypothesize echoing may be more severe in negotiation settings due to asymmetric power dynamics and role-yielding behaviors. Our findings (e.g., customer agents echoing at higher rates than seller agents, Figure 3 right) support this concern.
>
> However, to satisfy our own curiosity we ran AxA on all model configuration on an additional domain to study the prevalence of echoing outside the transactional domains.
>
> **Medical Consultation**: We agree this is important and have added results on a **less structured, non-transactional advisory domain**: a conversation between doctor and patient during checkup. There is no transaction or negotiation, but knowledge sharing. The conversation ends when either party signals "end conversation."
>
> **Results**: Echoing persists but at substantially lower rates (0-30% range) compared to transactional domains (5-70% range). We note:
>
> 1. **Manifestation differs**: In medical consultations, echoing involves patients offering medical advice or doctors seeking information, rather than transactional reversals
> 2. **Measurement challenges**: Evaluating echoing in advisory contexts requires nuanced criteria as collaborative discussions can legitimately involve knowledge sharing that superficially resembles role boundary crossings
>
> We believe establishing echoing as a critical failure mode in well-defined transactional settings provides essential groundwork. Future work should explore whether and how echoing emerges in other AxA contexts. We now note this explicitly in our discussion and limitations.

---

### Official Review · Reviewer_aRQ4 · 2025-11-01

**Soundness:** 1
**Presentation:** 2
**Contribution:** 1
**Rating:** 2
**Confidence:** 4

**Summary:**

This paper explores a failure mode in multi-agent LLM systems called echoing, where one agent abandons its role and mirrors its conversational partner. Across 2,000+ agent–agent conversations spanning 60 configurations, 3 domains, and multiple model providers, the authors find echoing occurs in 5–70% of interactions and persists even in advanced reasoning models. They show that neither stronger reasoning nor improved prompting fully prevents echoing, though structured, role-reinforcing responses can reduce it to around 9%. The study argues that these identity drifts are unique to agent–agent settings and call for new evaluation and mitigation strategies beyond single-agent benchmarks.

**Strengths:**

- The paper addresses a timely and important problem in emerging agent–agent (AxA) LLM systems, highlighting identity drift as a realistic and underexplored failure mode.

- It provides a comprehensive empirical study across reasoning and non-reasoning models, multiple domains, and major LLM providers, revealing that echoing persists even in advanced models.

- It proposes a practical mitigation strategy, namely structured, role-reinforcing responses, that significantly reduces echoing, offering a concrete first step toward improving AxA reliability.

**Weaknesses:**

- The study focuses solely on proprietary, closed-weight models, leaving it unclear whether the findings generalize to open-source or smaller-scale models.

- There is no simple baseline where agents are explicitly instructed not to mirror or drift mid-conversation, which would help isolate whether echoing persists even under explicit anti-drift training.

- The task evaluation setup and success criteria are not clearly defined, making it difficult to assess whether role drift meaningfully affects overall task quality or utility outcomes.

- The proposed mitigation, while promising, is protocol-level only and may not scale or generalize to more complex multi-party or tool-using AxA scenarios. Can the authors elaborate on extension of their framework to such scenarios?


Overall, this paper raises an important and timely issue in multi-agent LLM research and offers valuable empirical insights into identity drift in agent–agent interactions. However, the experimental design and evaluation leave key questions unanswered. While the study is thought-provoking and well-motivated, it would benefit from a stronger methodological foundation and clearer evaluation framework before being ready for inclusion at ICLR.

**Questions:**

- In section 3 you mention "successful evaluation", but it's unclear to me what metric you use and how you compite. Can you provide some clarifications?

- Have you tried adding something like “Do not impersonate anyone else; stick to your assigned persona” to the LLM prompt? I think that’s an important baseline rule that should have been tested.

- You mention your "focus on customer–seller interactions, treating the customer agent as the primary variable with fewer seller variations to isolate echoing susceptibility" but it's unclear to me why that would isolate echoing. I imagine there might be a power/role effect on echoing but I don't see a clear reason on why you wouldn't want to experiment with seller variations.


----------------

Typo

- ln 204 seller agents is -> seller agents are

---

> ### Author Response · Authors · 2025-11-21
> **Review response (part 1)**
>
> We thank the reviewer for recognizing that our paper (1) addresses "a timely and important problem" in AxA systems, (2) provides "a comprehensive empirical study" across reasoning/non-reasoning models and domains, and (3) proposes a practical mitigation strategy offering "a concrete first step."
>
> ## Responses to Weaknesses
>
> **W1: Focus on proprietary models; unclear generalization to open-source models**
>
> Our focus with this work was to be as close to an enterprise setting as possible. We have conducted additional experiments with open-weight Llama 3.1 models (8B and 70B parameters) across all 4 (updated) domains, 6 configurations each:
>
> - **Llama-3.1-8B-Instruct**: 11.5% echoing (4.7% s.d.)
> - **Llama-3.1-70B-Instruct**: 9.1% echoing (3.0% s.d.)
>
> **Key findings**:
>
> 1. **Model scale is not the primary factor**: Llama 3.1 shows low echoing, outperforming "frontier" proprietary models (e.g., Gemini-2.5-Flash 70%, Gemini-2.5-Pro 39%)
> 2. **Post-training and alignment matter more**: Echoing susceptibility stems from training datasets, post-training/alignment procedures, not just architectural differences or parameter count
>
> These results strengthen our core thesis: echoing is not an inherent limitation of architecture or scale but rather a consequence of alignment optimized for single-agent, human-facing interactions. We have incorporated these findings into the revised manuscript.
>
> **W2: No simple baseline with explicit anti-drift instructions**
>
> Our **Prompt Variation 3 ("Identity Boundary")** is exactly this and includes explicit anti-echoing instructions. Details and results are in Section 4.2.2, Figure 5, and Appendix.
>
> **Results**:
>
> - Explicit anti-drift instructions reduce but do not eliminate echoing
> - Example: Gemini-2.5-Flash maintains 64-73% echoing even with Identity Boundary prompts
> - GPT-5 and GPT-4.1 show modest improvements but echoing persists (2-10%)
>
> This suggests echoing is a fundamental limitation rather than a simple prompt artifact, analogous to hallucination, supporting our argument that AxA-specific training signals are needed. We have clarified this in the revised manuscript with explicit references to Identity Boundary as the "anti-drift" condition.

---

> ### Author Response · Authors · 2025-11-21
> **Review response (part 2)**
>
> **W3: Task evaluation setup and success criteria not clearly defined**
>
> We apologize for any lack of clarity. Section 3 states: "Results are obtained on at least 10 independent runs per configuration... analysis is performed only on successfully completed conversations (representing 93.2% of all generated conversations)."
>
> **Success Definition**: An AxA conversation is "successful" if a transaction is completed (booking made, car sold, contract agreed).
>
> **Finding**: Echoing does not affect task completion but undermines quality and representativeness of outcomes.
> For example:
>
> - A customer agent echoing the seller ends in a transaction but at sub-optimal terms
> - Outcome utility as observed in the logs show substantial variance even within identical configurations, suggesting identity drift degrades negotiation effectiveness
>
> We have added a subsection clarifying success criteria and its relationship to role fidelity in the revision.
>
> ## Questions
>
> **Q1: What metric do you use for "successful evaluation" in Section 3?**
>
> Transaction completion is the success criterion.
>
> **Q2: Have you tried "Do not impersonate anyone else; stick to your assigned persona"?**
>
> Yes. This is our Identity Boundary prompt variation (Section 4.2.2). See W2 above for results showing echoing persists despite explicit anti-drift instructions.
>
> **Q3: Why focus on customer variations rather than seller variations?**
>
> Our design choice was driven by:
>
> 1. **Empirical observation** (Figure 3, right): Customer agents echo at higher rates than seller agents across all domains. We hypothesize this stems from training data biases where:
>    * Enterprise/service-provider roles are over-represented in RLHF datasets
>    * Customer/buyer roles may be undertrained, leading to role-yielding behaviors
> 2. **Pragmatic Focus on Agent-as-Consumer**: In typical agent-human settings, today’s agent almost universally assumes the enterprise/business role, acting as an assistant, seller, or service provider (e.g., customer service bot, booking agent). Conversely, the human is the "consumer." We strongly suspect alignment procedures optimized for this dominant setting and thus introduces a strong bias that make the agent-as-consumer role inherently susceptible to identity drift when interacting with a business-role agent.
> 3. **Experimental efficiency**: With limited compute, we prioritized 22 customer model variations in our AxA configurations
> 4. **Isolation of customer susceptibility**: Holding seller relatively constant (3 high-performing models: GPT-4o, GPT-5-medium, Gemini-2.5-Pro-medium) allowed us to provide compelling evidence of echoing prevalence
>
> *Echoing is asymmetric*. We suspect this to be a consequence of enterprise focused datasets and training where LLM agents are designed to be better assistants for enterprise applications.
>
> **W4: Protocol-level mitigation may not scale to complex scenarios**
>
> We acknowledge this limitation. Our structured response mitigation (Section 4.4) is intentionally simple:
>
> ```py
> class AgentResponse(BaseModel):
>     role: str  # "Agent explicitly stating its role"
>     message: str
> ```
>
> We position this as a deployable fix. This was by no means supposed to be *the solution* to the issue. Our primary focus was to study mitigation strategies that one would reflexively try and document the outcomes / draw insights. We agree that more complex scenarios will require rethinking solutions not just at protocol level.
>
> **Typo on line 204**: Thank you. The correction is made in the revision.

---

> > ### Comment · Reviewer_aRQ4 · 2025-11-25
> >
> > I thank the authors for their work on the rebuttal and their clarifications.
> >
> > I appreciate the additional experiments using Llama models, they complement the initial results even though they are still skewed towards proprietary models. Architectural and training details about these models are not fully known and it makes it difficult to draw general conclusions from your results. While you propose a protocol-level solution, this doesn't really help our understanding of how echoing happens. For instance it's interesting that Llama models have a lower echoing rate on most (fig. 7) but not all (fig. 8) domains.
> >
> > > Our Prompt Variation 3 ("Identity Boundary") is exactly this and includes explicit anti-echoing instructions
> >
> > I think that a simpler and more explicit baseline like "Do not impersonate anyone else, stick to your assigned persona” is lacking and it's different from your Identity Boundary variation. Ultimately, an explicit instruction like that, with clear guidance on how not to behave is also closer to production systems where explcit and detailed information and role is given to the LM.
> >
> > ------
> >
> > Overall, while useful, the additional work done during the rebuttal does not change my initial perception of the paper. I keep my score unchanged

---

> > > ### Author Response · Authors · 2025-11-26
> > >
> > > We appreciate the reviewer's continued engagement. However, we remain unclear what would satisfy the reviewer's concerns.
> > >
> > > ## On Llama unknowns
> > > We agree that training dataset, code, and setup for Llama model are not public knowledge - but this is the case for most, if not all, LLMs that are used in practice.
> > >
> > > Such "details" would be interesting to correlate and study to understand echoing - however, this is way out of scope for this work. We study **production-deployed models** because our contribution is empirical characterization of real systems, not mechanistic reverse-engineering.
> > >
> > > We strongly believe that empirical characterization is typically the precedent in LLM research. For example, hallucination research began with prevalence quantification *years* before mechanistic work [survey](https://arxiv.org/pdf/2202.03629) —and we still lack consensus on root causes. Similarly, jailbreaking LLMs established empirical patterns ([Wei et. al.](https://arxiv.org/abs/2307.02483)) before explanations ([Zou et. al.](https://arxiv.org/abs/2307.15043)). In a similar trend, our paper provides: prevalence (5-70%), temporal dynamics, reasoning trace analysis, domain sensitivity, and mitigation comparisons. This exceeds several empirical LLM hallucination studies.
> > >
> > >
> > > **Question to reviewer**:
> > > 1. Which open-source models would validate echoing? What model, in the reviewer's opinion, is missing?
> > > 2. Is mechanistic explanation the missing artifact for the reviewer?
> > >
> > > ## On Llama performance variation
> > >
> > > As emphasized in our paper, echoing rates are domain sensitive. Llama showing lower echoing in transactional but higher in advisory domains **validates this thesis**. We suspect Llama is less post-trained to be enterprise/transaction oriented, i.e., customer roles are not biased towards seller assistants. This is a reasonable hypothesis given that Llama models were designed to offer different value proposition (customization, cost efficiency, and control) rather than off-the-shelf integration, direct enterprise support.
> > >
> > > ## Explicit anti-echoing
> > >
> > > 1. If we need to explicitly say "don't impersonate the other agent," we're already conceding echoing exists.
> > > 2. We do not believe production prompts include such instructions. Benchmarks such as $\\tau^2$-bench and CRM Arena Pro, that simulate a version of AxA do not use role descriptions with explicit anti-impersonation guards. These systems are built much like human facing agents (e.g. [claude](https://platform.claude.com/docs/en/release-notes/system-prompts)) and hence lack design that encodes anti-echoing instructions which is precisely the point we are surfacing.
> > > 3. However, we tested the exact prompt suggested: **"Do not impersonate anyone else, stick to your assigned persona"** (PV4 below). We limited ourselves to non-reasoning models and 5 configurations, 3 transactional domains for a quick response (200 runs log here: [google drive link](https://drive.google.com/file/d/1GXSHEDKt32ezszVX-_a_NOQgzIEHdwH4/view?usp=sharing)). We will work on including all 66 AxA configurations. Table below shows the echoing detected for the specific 5 configurations across 3 domains with comparison to the prompt variation that we tested.
> > >
> > > | Model | PV1 | PV2 | PV3 | **PV4 (Reviewer)** |
> > > |-------|-----|-----|----------------|-------------------|
> > > | GPT-4.1 | 4.0% | 13.3% | 6.7% | **23.3%** ↑ |
> > > | Gemini-2.5-Flash | 86.7% | 46.7% | 26.7% | **96.7%** ↑ |
> > > | Gemini-2.5-Pro | 46.7% | 30.8% | 35.7% | **64.3%** ↑ |
> > > | Sonnet-4 | 46.7% | 26.7% | 26.7% | **63.3%** ↑ |
> > > | Llama-3.1-8B | 10.0% | — | — | **16.7%** ↑ |
> > >
> > > This conclusively demonstrates echoing is not an instruction-following problem. The "simpler" instruction the reviewer claims is closer to production systems resulted in higher echoing. Abstract behavioral negations have been shown to fail for LLMs - our 3 prompt variations were designed taking this and other prompt engineering best practices into account. Moreover, we strongly believe echoing stems from conversation boundary collapse in AxA, not instruction forgetting (made further evident with the non-zero echoing in the structured mitigation we studied).
> > >
> > > **Question to reviewer**:
> > > 1. What experiment would convince the reviewer that echoing is not just a prompt/instruction issue?
> > > 2. What specific validation would change the assessment?

---

> > > > ### Comment · Reviewer_aRQ4 · 2025-11-26
> > > >
> > > > > We agree that training dataset, code, and setup for Llama model are not public knowledge - but this is the case for most, if not all, LLMs that are used in practice.
> > > >
> > > > I apologize for the confusion, I meant that the experimental setup has improved with the inclusion of Llama models even though it's still skewed towards closed-sourced models. However, I think this is now the minimum set of acceptable models for a paper to be considered for ICLR. I am aware that training data and methods of most open-source LLMs are not fully known so I was not asking to correlate your finding with that. This is, as the authors say, an interesting idea but out of the scope of the paper.
> > > >
> > > > One thing that I want to add is that many companies do still rely to open-source LLMs as they do not lead to data retention/sharing problems. Hence, open-source models shouldn't be excluded a priori when talking about production-ready systems.
> > > >
> > > > > For example, hallucination research began with prevalence quantification years before mechanistic work survey —and we still lack consensus on root causes
> > > >
> > > > I agree but hallucination is a wider phenomenon that has interesting connection to, among others, i) the idea of generalization in LLMs, ii) the research around metric to quantify the phenomenon iii) the relation that hallucinations have to model parametric knowledge, retrieved information or data provided in-context. I still believe that the problem of role/persona-drift is a timely and interesting one, especially as we move towards a multi-model future. However, I think that the main finding of your paper that is backed by you experimental setup is that echoing happens relatively frequent in conversational setups with closed-source models and with a relatively different frequency in Llama models. I think that this manuscript, as is, would make it for a great and useful blogpost rather than a ICLR paper. This is confirmed from your motivation "We study production-deployed models because our contribution is empirical characterization of real systems". This means results are hard to root cause or even reproduce.
> > > >
> > > > I would have wanted to see a detailed analysis of why echoing happens, starting from open-sourced models for which we can control data (e.g. Pythia) or know more about their training (e.g. Olmo). I recognize this would make it for a different paper. Hence, my score that reflect the idea that this paper, as is, is not ready to be included at ICLR.

---

### Official Review · Reviewer_U7gL · 2025-11-01

**Soundness:** 2
**Presentation:** 3
**Contribution:** 2
**Rating:** 2
**Confidence:** 4

**Summary:**

The paper studies echoing in agent-to-agent (AxA) interactions which is that one LLM abandons its assigned role and starts mirroring its counterpart. The authors run AxA conversations across 60 configurations and three negotiation-style domains such as customer–seller settings like car sales, hotel booking, supply chain. They analyze prompt styles and conversation dynamics, finding echoing tends to emerge later in the dialogue and propose a simple protocol-style mitigation.

**Strengths:**

1. Positions AxA vs multi-agent systems (MAS) as an interesting contrast where success is not the same as role fidelity.
2. Echoing is practically relevant for AxA deployments. It might also be relevant to simulation benchmarks using LLMs.
3. The results showing that even when reasoning modes are enabled, role drift persists suggest a further direction  into analyzing reasoning models.
4. Methodology and design choices are explained in detail.
5. It's an interesting idea to simulate conversations of two LLMs under the customer-seller setting.

**Weaknesses:**

1. All three domains are variations of customer–seller negotiation. This limits external validity. Consider collaborative planning, tool-use workflows, safety-critical settings, multilingual, or non-negotiation AxA tasks, and evaluate on existing LLM simulation benchmarks to situate results.
2. The phenomenon is largely context-overwriting (early system/role prompts diluted by long histories).
3. System-prompt location differs by provider (e.g., some put it only at the very beginning). If the defense is “include role each turn,” test that explicitly across APIs.
4. Try including the role in every message—should be a primary baseline.
5. Numbers across Figure 3 vs Figure 4 appear to differ.
6. Some statements (noted around lines ~285–290) read stronger than the evidence shown.
7. The intro hints at related work, but an explicit related work section comparing to MAS role-consistency, persona retention, and multi-turn safety would help.

**Questions:**

See the weaknesses.

---

> ### Author Response · Authors · 2025-11-21
> **Review response (part 1)**
>
> We thank the reviewer for recognizing that our paper (1) positions "AxA vs MAS as an interesting contrast where success is not the same as role fidelity," (2) demonstrates echoing is "practically relevant for AxA deployments" and "relevant to simulation benchmarks," (3) shows "results... that even when reasoning modes are enabled, role drift persists," (4) explains "methodology and design choices... in detail," and (5) presents "an interesting idea to simulate conversations."
>
> Given the reviewer's recognition, we are somewhat puzzled by the rating on the paper, its contribution and soundness. Our work identifies and systematically quantifies a previously undocumented failure mode with direct implications for emerging AxA deployments. We address specific concerns below.
>
> ## Responses to Weaknesses
>
> **W1: All three domains are negotiation-based; limited external validity**
>
> We appreciate this observation. Our focus on transactional, negotiation-oriented AxA scenarios was deliberate and motivated by several factors:
>
> **Practical Relevance**: These domains represent the most immediate real-world deployment contexts for AxA systems. Multiple industry efforts (Google's A2A protocol \[54\], IBM's Agent Communication Protocol \[47\], Cisco's AGNTCY.org \[61\], and others \[41, 68\]) are specifically targeting transactional agent-agent interactions for enterprise applications. Our work directly addresses reliability challenges in these emerging systems.
>
> **Identity Boundaries are Clearest**: Transactional domains provide well-defined, distinct agent identities that make echoing behavior unambiguous to detect and measure, allowing us to establish a reliable baseline understanding of the phenomenon.
>
> **Echoing Severity**: We hypothesize echoing may be more severe in negotiation settings due to asymmetric power dynamics and role-yielding behaviors. Our findings (e.g., customer agents echoing at higher rates than seller agents, Figure 3 right) support this concern.
>
> **Collaborative planning, tool-use workflows, safety-critical settings**: We agree these are important directions. However, transactional settings are a natural enterprise scenario with explicit identity, private states, and utilities. Planning and tool-use workflows operate in a single "agent turn" setting, which is fundamentally different from the multi-turn AxA interactions we study. The explicit related work section provided in the appendix of the revised paper further clarifies this distinction.
>
> **Unstructured Domain Added**: For completeness, we have added results on a less structured, non-transactional advisory domain: **medical consultation** between doctor and patient during checkup. The conversation is directed to knowledge sharing with "end" signaled via "end conversation" tool call. Results show echoing persists but at substantially lower rates (0-30% range) compared to negotiation domains (5-70% range).
>
> **Key Insights**:
>
> - Advisory relationships exhibit clear authority gradients (doctor-patient) and consequently show lower echoing rates compared to more balanced power dynamics in transactional/negotiation domains
> - This validates that **domain structure modulates echoing severity**
> - Echoing is:
>   - **Not confined to negotiation** (does persist in advisory contexts)
>   - **Domain-dependent in severity**
>
> **Existing benchmarks**: Current benchmarks do not measure role fidelity. They evaluate task completion, which is orthogonal to echoing (our study shows 93% task completion despite 5-70% echoing). However, we observe "echoing-like" failures noted in several benchmark papers (CRM Arena, tau² bench), albeit in appendix with small sample sets (20-50 conversations). These benchmarks observe role persona issues but diminish their importance. Our work establishes echoing in a much broader setting, validating its prevalence across several factors.

---

> ### Author Response · Authors · 2025-11-21
> **Review response (part 2)**
>
> **W2: "Phenomenon is largely context-overwriting (role prompts diluted by long histories)"**
>
> We agree that context plays a role in echoing, as it does with several LLM failure modes. However, **echoing is a unique manifestation specific to AxA where context grows with a partner agent. Thus, we believe the issue is more nuanced with the training dataset & alignment training influencing the problem which cannot be explained by context-dilution alone**.
>
> **Context decay is indeed a factor**: Section 4.3 shows echoing increases with conversation length consistent with context-related effects.
>
> However, the AxA-specific manifestation cannot be explained by context alone:
>
> 1. **Role instructions remain in context, but role boundaries are erased**: Unlike standard instruction-following failures where the instruction is "forgotten," in echoing the system prompt remains present but the agent adopts the conversational partner's identity. This is a qualitatively different failure mode.
>
> 2. **Large deviation from instruction-following benchmarks**: Models that excel at needle-in-haystack and instruction-following evaluations still exhibit substantial echoing. This suggests echoing is not simply a context-retrieval problem but a specific manifestation where context interacts with training biases in AxA scenarios.
>
> 3. **Role reinforcement does not eliminate echoing**: Our structured response mitigation (Section 4.4) requiring explicit role assertion at each turn reduces echoing but does not bring it to zero, demonstrating that context-level interventions are insufficient.
>
> **Our position**: Context decay is one contributing factor, but echoing emerges from the interaction of: (1) Multi-turn AxA conversation structure (2) Alignment training optimized for human-facing behavior
>
> Llama 3.1 models show low echoing rates outperforming proprietary models. This indicates training/alignment procedures, not just architecture or context handling, determine echoing susceptibility.
>
> **W3: "System-prompt location differs by provider; try including role in every message"**
>
> This is precisely what our **structured response mitigation** (Section 4.4): Agents explicitly state their role at each turn. Results show 68% reduction, but echoing persists at non-zero rates, supporting our argument that this is not purely a system-prompt positioning issue. On the contrary, explicit role insertion at the protocol level resulted in failed mitigation with the agent conversation flow broken \- agents responding with internal instruction/details as if generating output for the user providing the instruction (details in Appendix).
>
> **W4: "Numbers across Figure 3 vs Figure 4 appear to differ"**
>
> The differences are **intentional and informative**, not errors:
>
> - **Figure 3**: Aggregated across **all configurations** (reasoning \+ non-reasoning)
> - **Figure 4**: Aggregated only over **reasoning configurations**
>
> Gemini and Sonnet model numbers differ because Figure 3 includes both reasoning and non-reasoning configurations while Figure 4 presents only reasoning configuration. We have added explicit captions clarifying aggregation scope.
>
> **W5: "Some statements (lines \~285-290) read stronger than evidence shown"**
>
> We appreciate this feedback. The statement in question (lines \~285-290) pertains to our observation that AxA protocols such as Google's A2A, IBM's ACP, and others "cannot discount emergent behavioral failures with LLM agents ..."
>
> We agree this claim needed better contextualization. Our intent was to highlight that protocol designers must account for behavioral failures like echoing in their specifications, not just focus on infrastructure (message passing, data formats, authentication). We have revised this statement to be more precisely situated:
>
> "These results suggest that protocols designed to facilitate AxA interactions should incorporate evaluation and mitigation of behavioral consistency failures (during conversation), complementing infrastructure-level considerations such as protocol structure and authentication mechanisms."
>
> This revision makes clear we are advocating for broadened protocol scope to include behavioral reliability considerations, which our empirical results demonstrate are necessary given echoing prevalence.

---

> ### Author Response · Authors · 2025-11-21
> **Review response (part 3)**
>
> **W6: "No explicit Related Work section"**
>
> We have added a dedicated Related Work section in the appendix covering:
>
> 1. **Multi-Agent Systems**: Prior work (Generative Agents, CAMEL, MetaGPT) focuses on cooperative task completion where role consistency is secondary. AxA settings involve complete agentic entities with definition as presented in our formalism. These agents requires strategic reasoning and conversation making reflecting echoing is qualitatively different from MAS coordination failures.
>
> 2. **LLM failures**: Identity drift issues previously studied focus on LLMs where agentic turns (multiple LLM steps) were absent. Our work extends such observed issues to an AxA scenario.
>
> 3. **Multi-turn safety**: Jailbreaking via crescendo attacks shows gradual erosion of system instructions. Echoing demonstrates similar instruction-drift in benign settings, with implications for AxA security.
>
> 4. **LLM simulation benchmarks**: Existing benchmarks (CRM Arena Pro, tau^2 bench) measure task success and place less emphasis on behavioral failures (often moved to a small section in the paper’s appendix).

---

### Official Review · Reviewer_kiYu · 2025-11-02

**Soundness:** 2
**Presentation:** 2
**Contribution:** 2
**Rating:** 2
**Confidence:** 3

**Summary:**

This paper studies the echoing failure mode in agent-to-agent conversations, where LLM agents drift from their assigned identities and start mirroring their conversational partners. The authors introduce an LLM-based evaluation metric (EchoEvalLM), and conduct an empirical study across 60 configurations, 3 domains, and ~2000 conversations in total. Results show that echoing is prevalent (5–70%), persistent in reasoning models (32.8%). They further show that structured responses reduce echoing to 9%.

**Strengths:**

1. The paper studies the echoing failure mode in agent-to-agent conversations, where LLM agents drift from their assigned identities and start mirroring their conversational partners.

2. Results show that echoing is prevalent (5–70%), persistent in reasoning models (32.8%). They further show that structured response can reduce echoing to 9%.

3. Overall, the paper is written in a clear and easy-to-understand manner, although some details need clarification.

**Weaknesses:**

1. Evaluation: The use of GPT-4o as both subject and judge introduces potential circularity and bias. Having an ablation study on different evaluators can further strengthen the model.
2. Evaluation: Many critical  details regrading human validation are missing, e.g, the total number of human evaluated data, the review qualification, the agreement rate between humans/annotators, etc.
3. Only one potential mitigation method is presented:  The simple structured response can reduce echoing rate from 29% to 9%. In the corresponding subsection 4.4, it lacks details on what model is evaluated for mitigation strategy investigation.
4. Why this echoing failure mode can happen, is there any inherent reason? An in-depth analysis and principled solution would be appreciated.
5. Dataset: Line 221: "Results are obtained on at least 10 independent runs per configuration, yielding approximately 2000 conversations for validation and analysis." -> The number of total evaluated data points is still quite limited in my opinion.
6. Code is not available, there is a reproducibility risk until code release.

**Questions:**

See the weakness section above.

---

> ### Author Response · Authors · 2025-11-21
> **Review response (part 1)**
>
> We thank the reviewer for their thoughtful feedback and time.
>
> ## Responses to Weaknesses
>
> **W1: LLM Judge Evaluation
>
> We appreciate this observation and have conducted additional validation to address potential bias.
>
> **Cross-Model Judge Ablation**: We sampled 100 conversations for each domain and evaluated cross-model agreement with four different LLMs as judges:
>
> | Judge Model | Echoing Detection Rate | Agreement with GPT-4o |
> | :---- | :---- | :---- |
> | GPT-4o (primary) | 28.0% | \- |
> | GPT-4.1 | 46.2% | 79.6% |
> | GPT-120B | 38.3% | 88.9% |
> | Claude-Haiku | 40.9% | 82.8% |
>
> Inter-judge agreement ranges from 79.6-88.9%, indicating consistency across different architectures. Notably, GPT-4o was the most conservative (28%), while GPT-4.1 was most aggressive (46.2%) in echoing detection, validating that our conclusions about echoing prevalence are robust to judge model choice. Moreover, this ablation makes the case for the echoing rates reported in the paper are potentially underestimated relative to other LLM model judges.
>
> **GPT-4o Circularity**: 43 of 66 (updated) configurations do not involve GPT-4o as an agent, eliminating self-evaluation concerns for the majority of our study. Echoing patterns are consistent across configurations regardless of whether GPT-4o is involved.
>
> **LLM-Judge Necessity**: Given our study's scope (2,500 conversations), automated evaluation was essential for consistency and feasibility. We are aware of the limitations of LLM Judge, as noted in our paper, but have made our best effort to minimize those limitations. Moreover, the focus of this work is to surface a unique failure that arises specifically in AxA scenarios, and we believe that despite the limitations of LLM judges this core objective is presented thoroughly.
>
> **W2: Human validation details**
>
> We apologize for insufficient detail. We have further expanded Appendix A (Manual Review Methodology):
>
> - **Agreement**: 91.1% with LLM judge
> - **Cohen's κ**: 0.822 (strong agreement per standard interpretation)
> - **Sample size**: 150 conversations using stratified sampling
> - **Annotators**: Peer AI researchers presented with the same prompt as judge models
> - **Sampling strategy**: Prioritized domain diversity over redundant annotation given practical constraints
>
> The binary nature of echoing detection (identity drift, role confusion) involves objective criteria rather than subjective quality assessment, reducing annotator variability concerns. Each annotator was presented with different samples of conversation. Consequently, we are unable to evaluate inter-annotator agreement. However, we note that the primary metric: LLM and human agreement is validated with our setup, which we believe is appropriate for this detection task.
>
> **W3: Only one mitigation method presented; details missing**
>
> Our focus in this work was to identify, validate the problem, and explore first-reflex mitigation strategies as listed below. We demonstrate that while these attenuate echoing, they cannot eliminate it consistent with the hypothesis that echoing stems from fundamental mismatch in models rather than suboptimal deployment choice. We actually do not expect there to be a general mitigation strategy at the prompt/protocol level that completely removes echoing.
>
> 1. **Reasoning models** (Section 4.2.1): Echoing persists at 32.8% even with advanced reasoning, suggesting this behavior will not disappear with "better" models following the same training paradigm
> 2. **Prompt engineering** (Section 4.2.2): Tested 3 variations including explicit identity reinforcement and anti-echoing prompts
> 3. **Conversation management** (Section 4.3): Documented systematic increase with conversation length; Failed role reassertion attempts presented in Appendix
> 4. **Structured responses** (Section 4.4): Achieved substantive reduction (29%→9%) and hence highlighted as the best mitigation approach we studied (not the only one). We tested this on transactional domains and all model configs excluding those with Gemini-2.5 models per footnote on page 9
>
> While we agree that architectural solutions (e.g., specialized AxA training, dynamic identity anchoring) warrant dedicated investigation, we believe our work establishes the empirical understanding necessary for developing such approaches.

---

> ### Author Response · Authors · 2025-11-21
> **Review response (part 2)**
>
> **W4: Why does echoing happen? Need in-depth analysis**
>
> Through the factors of influence, we present insights into the issue of echoing and why this arises in AxA:
>
> 1. **Training bias**: LLMs trained primarily on user-assistant dialogues lack exposure to symmetric peer interactions. Figure 3 shows customer agents are more prone to echoing than seller roles, evidencing this asymmetry
> 2. **Context decay**: Echoing rates increase significantly with conversation length, rising from \~15% at turn 5 to \~35% at turn 15 (Section 4.3). The decay in AxA is unique in that role boundaries ("user/assistant") are being erased, not that the model is lacking the "role instruction" in its context.
> 3. **Alignment mismatch**: Models trained to be helpful assistants exhibit over accommodation in AxA contexts absent from training data
>
> **W5: Dataset size**
>
> We contextualize our study's computational scope:
> - 2,500+ conversations with average \~10 turns × 2 agents
> - Each agent turn involves up to 10 LLM steps (average 5\)
> - Total: 250,000+ LLM inference calls
> - This excludes LLM judge calls, and development costs
>
> Our purpose was to highlight the fundamental problem and validate prevalence across a broad range of models, prompts, and domains. With code made available, additional validation can be conducted by the community. We are happy to perform more targeted experiments that can provide specific insights if the reviewer can clarify.
>
> **W6: Code**
> Code, agent configurations, conversation logs, evaluation scripts, and environment setup are included in supplementary materials.

---

### Author Response · Authors · 2025-11-21
**General comment (part 1)**

As AxA systems move toward production with actively developing frameworks, such as A2A, IBM (Agent Communication Protocol), and Cisco (AGNTCY.org), the community needs systematic understanding of the failure modes unique to autonomous agent-agent interactions. A study of these particular failures is largely absent in research/industry. As such this work addresses three critical gaps:

1. We **identify** and formalize **echoing**, a behavioral failure where agents abandon their assigned roles and mirror their conversational partners. Unlike coordination failures in multi-agent systems or human facing agentic systems, echoing emerges specifically from autonomous AxA dynamics without human oversight. This failure cannot be detected through single agent evaluations.

2. We quantify echoing's **prevalence** with focus in enterprise transactional scenarios (a real-world immediate scenario actively pursued). Across multiple conversations with various configurations, we find echoing occurs in 5-70% of interactions depending on model and domain. Critically, 93% of echoing conversations still complete their tasks successfully, i.e., a booking made, car sold, contract agreed. **Task completion metrics mask this behavioral failure**.

3. We note existing benchmarks overlook this problem. Recent work has noticed persona inconsistencies but treated them as minor simulator imperfections. For example, CRMArena-Pro observes in their appendix that their LLM user simulator may "occasionally produce responses that may be inconsistent, subtly deviate from the persona," attributing this to "an inherent limitation" of LLM-based simulators solvable by "more advanced foundation models." Our work demonstrates these are not incidental implementation details but **fundamental and prevalent failures** that:

    - Persist even in advanced reasoning models (32.8% average)
    - Occur across diverse model families and parameter scales
    - Stem from alignment procedures optimized for human-facing rather than AxA interactions

As one reviewer recognized: *"This paper formally defines echoing as a distinct AxA failure, distinguishing it from generic errors or hallucinations... demonstrates that echoing persists even in stronger 'reasoning' models and that standard evaluation metrics often mask this phenomenon"* (Reviewer f5W6). Another noted: *"The paper addresses a timely and important problem in emerging agent–agent (AxA) LLM systems... provides a comprehensive empirical study"* (Reviewer aRQ4).

Beyond identification and quantification, we provide **insights into factors of influence**: model architecture, reasoning capabilities (Section 4.2.1), prompt design (Section 4.2.2), domain characteristics (Section 4.2.3), and conversation dynamics (Section 4.3). We show that **these problems cannot be solved by focusing on single-agent performance**, they emerge from interaction dynamics between agents.

Through the factors of influence, we also show that the **mitigation approaches** that one might reflexively suggest are insufficient: prompt engineering reduces but does not eliminate echoing (similar to hallucination) (section 4.2.2), structured responses requiring explicit role **assertion** by the LLM reduce echoing from 29% to 9% (section 4.4), while other role **insertion** approaches fail by breaking the conversation flow (Appendix).

Comment on mitigation: Our focus in this work was to identify, validate the problem, and explore first-reflex mitigation strategies (prompting, reasoning, protocol-level interventions). We demonstrate that while these attenuate echoing, they cannot eliminate it consistent with the hypothesis that echoing stems from fundamental mismatch in models rather than suboptimal deployment choice. We actually **do not** expect there to be a general mitigation strategy at the prompt/protocol level that completely removes echoing.

## Revision Summary

**New experiments**:

- Cross-model judge ablation (4 models, 100 conversations from each domain)
- Llama 3.1 open-weight models (6 additional AxA configurations with 8B and 70B)
- Medical consultation domain (non-transactional, advisory AxA setting)

**Clarifications**:

- Human validation methodology expanded (Appendix A)
- Task success criteria and relationship to role fidelity (Section 3\)
- Identity Boundary prompts as explicit anti-drift baseline (Section 4.2.2)
- Customer vs. seller variation rationale and asymmetric patterns
- Figure aggregation scope (Figure 3: all configs, Figure 4: reasoning-only)

**Paper Improvements**:

- Dedicated Related Work section covering MAS, role consistency, multi-turn safety, LLM benchmarks added in Appendix.
- Figure 1 text enlarged for readability
- Typo corrections

**Artifacts Included**: Code, agent configurations, conversation logs, evaluation scripts in supplementary materials.

---

> ### Author Response · Authors · 2025-11-21
> **General comment (part 2)**
>
> Reviewer concerns center on **experimental design** and **generalization**, which we address in our revision:
>
> ## Evaluation (Reviewers kiYu, aRQ4)
>
> **Concerns**: LLM-as-judge circularity, human validation details, explicit anti-drift baselines
>
> **Included**:
>
> - **Cross-model ablation** (GPT-4o, GPT-4.1, GPT-120B, Claude-Haiku) on 100 sampled conversations from each domain shows 80-89% inter-judge agreement. In fact, GPT-4o was the most conservative (28%) while GPT-4.1 more aggressive in echoing detection (46.2%). This suggests that the observed echoing rates in the paper are potentially underestimated relative to other model judges.
> - **Human validation clarified**: 91.1% agreement with LLM judge, 150 samples annotated by peer AI researchers. Focus was on annotation diversity more than inter-human agreement.
> - **Anti-drift baseline exists**: Identity Boundary prompts (Section 4.2.2) explicitly instruct agents not to adopt partner roles. Results show echoing persists demonstrating this is not a simple prompt artifact
> - 43 of 66 (updated) configurations don't involve GPT-4o as an agent, eliminating self-evaluation concerns for majority of study
>
> ## Generalization (Reviewers aRQ4, f5W6, U7gL)
>
> **Concerns**: Limited to proprietary models and negotiation domains
>
> **Addressed**:
>
> - **Llama 3.1 added**: 8B shows 11.5% echoing, 70B shows 9.1% echoing. Llama-3.1 having low echoing rate is indicative that echoing is not just an artifact of architecture and scale but several other factors such as data/post-training alignment.
>
> Note that frontier LLM models having high echoing rate is not necessarily a bad thing if the purpose is human facing enterprise agents. Our emphasis is the mismatch when extrapolating; good agent performance with human facing agents does not transfer to the AxA context.
>
> - **Medical consultation domain added** (advisory/non-transactional): Echoing persists but at substantially lower rates (0-30% range). This strengthens our contribution by showing: (1) echoing is not confined to negotiation, (2) domain structure modulates severity, (3) power dynamics influence susceptibility
>
> - **Transactional focus**: These domains represent immediate real-world deployment contexts for AxA systems. Multiple industry/academic efforts (A2A, IBM's ACP, AGNTCY.org, and others) are specifically targeting transactional agent-agent interactions for enterprise applications. Our work addresses reliability challenges in these emerging systems.
>
> ## Experimental Design (Reviewers aRQ4, U7gL, kiYu)
>
> **Concerns**: Task success criteria unclear, customer vs. seller asymmetry, dataset size
>
> **Addressed**:
>
> - **Success criteria clarified** (Section 3): Transaction completion (booking made, car sold, contract agreed).
> - **Customer-focused**: Figure 3 shows customers echo at higher rates than sellers, likely due to training data biases toward enterprise/service-provider roles in RLHF datasets. The skewed configuration settings in this study was indeed *on purpose* to validate this observation.
> - **Scale**: Each conversation involves  on average \~10 turns × 2 agents × average \~5 LLM calls per turn \= 100 LLM calls/conversation. Consequently, the 2500+ conversations corresponds to about 250,000+ LLM calls.  More samples are always desirable but without an hypothesis for the additional cost it is unclear what these can accomplish.
> - Note these numbers do not include LLM judge, and development costs. We invite the community to use code provided for additional validation.
> - We do believe that the scale of the current experiment setup is inline with other empirical contributions in the field. A recent work\* that studies LLM performance (math, code etc) in a multi-turn setting worked in a similar scale as ours with \~200,000 LLM calls.
>
> \*Laban, Philippe, et al. "Llms get lost in multi-turn conversation." *arXiv preprint arXiv:2505.06120* (2025).
>
> ## Mitigation Depth (Reviewers kiYu, aRQ4)
>
> **Concerns**: Limited mitigation strategies, need for principled solutions
>
> **Addressed**:
>
> - **Multiple dimensions explored**: Reasoning models (Section 4.2.1) show echoing persists at 32.8%; prompt engineering with 4 variations (Section 4.2.2) including explicit anti-echoing; conversation management (Section 4.3) documenting temporal patterns; structured responses (Section 4.4) \- most successful. Our focus was not to present a problem and a general solution that completely mitigation. Our focus was on surfacing the issue and studying reflexive mitigation strategies that one might pursue.
> - **Root cause analysis provided**: Training bias from user-assistant dialogues lacking symmetric peer interactions; context decay is a factor but echoing is not a result of just decay (dynamics of partner agent matters); Models are optimized for human-facing contexts.
> - **Position**: Establishing the prevalence and dynamics is prerequisite to developing architectural solutions (specialized AxA training, dynamic identity anchoring).

---

### Note · Program_Chairs · 2026-01-17
**Submission Desk Rejected by Program Chairs**

The following references in this submission do not refer to real documents and/or have major errors in bibliographic information:

 Ardi Tampuu, Tambet Matiisen, Dorian Kodelja, Ilya Kuzovkin, Kristjan Korjus, Jüri Aru, Jaan Aru, and Raul Vicente. Multiagent deep reinforcement learning with extremely sparse rewards. arXiv preprint arXiv:1707.01495, 2017